# Radio-transparent dipole antenna based on a metasurface cloak

Jason Soric[1,5], Younes Ra'di[2,5], Diego Farfan[1,2] & Andrea Alù [1,2,3,4 ✉]

Antenna technology is at the basis of ubiquitous wireless communication systems and sensors. Radiation is typically sustained by conduction currents flowing around resonant metallic objects that are optimized to enhance efficiency and bandwidth. However, resonant conductors are prone to large scattering of impinging waves, leading to challenges in crowded antenna environments due to blockage and distortion. Metasurface cloaks have been explored in the quest of addressing this challenge by reducing antenna scattering. However, metasurface-based designs have so far shown limited performance in terms of bandwidth, footprint and overall scattering reduction. Here we introduce a different route towards radio-transparent antennas, in which the cloak itself acts as the radiating element, drastically reducing the overall footprint while enhancing scattering suppression and bandwidth, without sacrificing other relevant radiation metrics compared to conventional antennas. This technique opens opportunities for cloaking technology, with promising features for crowded wireless communication platforms and noninvasive sensing.

[1] Department of Electrical and Computer Engineering, The University of Texas at Austin, Austin, TX 78712, USA. [2] Photonics Initiative, Advanced Science Research Center, City University of New York, New York, NY 10031, USA. [3] Physics Program, Graduate Center, City University of New York, New York, NY 10016, USA. [4] Department of Electrical Engineering, City College of The City University of New York, New York, NY 10031, USA. [5]These authors contributed equally: Jason Soric, Younes Ra'di. ✉email: aalu@gc.cuny.edu

High-density integrated wireless systems continue to grow at a very fast pace in our ever-connected world. Driven by increasingly overcrowded wireless channels, there is a burgeoning demand for co-located antennas and sensors operating in overlapping or disjoint frequency channels on the same physical platform. However, nearby antennas act as parasitic elements for each other, with negative impact on each other's radiation performance, including beam-squinting, blockage (shadowing), and gain distortion of neighboring radiation patterns. The parasitic effects concurrently act as detuning elements on the mutual input impedance[1]. A technique based on choked antenna arms has been explored to address these challenges in interleaved dual-band base station antenna arrays[2], but with inherent limitations on the antenna shape and performance.

In an effort to address these challenges, metasurfaces and metamaterials[3–16] have been explored as a way to eliminate antenna blockage and co-site interference reduction in a compact and scalable platform, exploiting the concept of cloaking and scattering cancellation[17–24]. In particular, metasurface cloaks based on hard surfaces[25,26], transmission lines[27–29], and scattering cancellation techniques[30–43] have been applied over the years to reduce antenna blockage in various scenarios. These approaches leverage the effectiveness of metasurface cloaks to suppress the scattering and shadows of conventional conductive targets, restoring the impinging wavefronts independent of the source location. However, resonant antennas have very strong scattering to start with over the entire frequency band of interest, making the goal of large scattering suppression, at the same time avoiding detuning, a challenging task. An additional significant disadvantage of these methods is the increased overall size of the cloaked antenna and its fabrication complexity. In particular, given that conventional antennas are conductive, a conventional metasurface cloak cannot be simply wrapped around the metal, but it needs to include a spacer with sufficient thickness dictated by the required bandwidth[34]. This feature leads to awkward designs that may not be easily scaled for mass production. The large footprint of metasurface cloaks over conventional antenna systems also implies polarization selectivity, effective operation over a limited range of incidence angles, and narrow operational bandwidth. These restrictions have so far hindered the broad practical impact of cloaking technology in wireless antenna and sensor systems.

Here, we explore an alternative route to enable radio-transparent antenna technology, which addresses all these challenges. We start from a metasurface cloak designed to suppress the scattering from a dielectric core. Given the limited scattering of nonresonant dielectric objects of subwavelength size, the cloak can make the overall scattering very low over a broad bandwidth. At the same time, given the non-conductive nature of the object, the cloak can be wrapped conformally to the object, or even embedded inside it, providing broadband scattering suppression and angular stability in a very compact design. Quite interestingly, the cloaking surface aimed at canceling the scattering of the dielectric core can actually be designed to also operate as an antenna over the desired frequency band. We demonstrate its operation in a commercial base station antenna panel, showing that it can meet all relevant metrics of performance in terms of radiation features, enabling an ideally suited broadband radio-transparent antenna.

## Results

### Theoretical analysis
In order to showcase the operation of the proposed concept, we consider a typical wireless communication scenario, in which antennas operating in different frequency bands are forced to operate in close proximity. A simplified

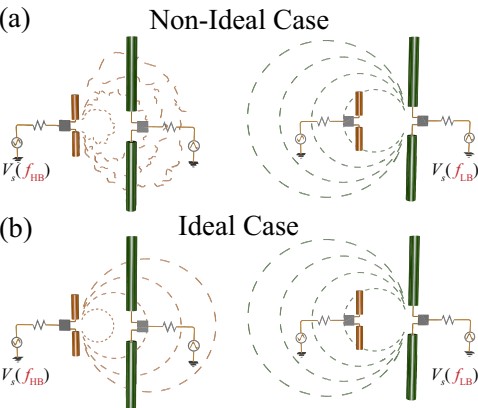

**Fig. 1 Radiation from closely located antennas. a** Non-ideal scenario in which a low-frequency antenna drastically affects the radiation at high frequencies; **b** Ideal scenario of a radio-transparent low-frequency antenna.

illustrative example, consistent with the typical geometry of a multi-band radio-base station antenna array, is shown in Fig. 1, where two dipole antennas of different size operating in different frequency bands are located close to each other. The smaller antenna operates in the higher frequency band ($f_{HB}$) and it does not significantly affect the radiation features of the larger antenna operating in the lower band ($f_{LB}$) given the small size. However, the larger antenna drastically deteriorates the radiation at $f_{HB}$, affecting its polarization, beam-squinting, and inducing undesired shadows in the radiation pattern [see Fig. 1a]. Our goal is to address this challenge by realizing a radio-transparent low-frequency antenna that offers minimal interference [see Fig. 1b]. Compared to other approaches, we demonstrate a broad bandwidth over which the antenna has very limited scattering signatures, with a design that can be straightforwardly extended to different antenna lengths and complex geometries within a simple and scalable fabrication process.

To analyze the conventional cloaking methods that have been employed to address these issues, and to also introduce our proposed approach, we start by investigating the multilayer cylindrical geometry shown in Fig. 2a. Equations (1) and (2) explicitly describe the fields in each region of the 2D dielectric cylinder illustrated in Fig. 2a[44]:

$$E_z = E_{z0}e^{jk_{zi}z}\sum_n j^{-n}e^{jn\phi}\begin{cases} x_{1n}J_n(k_{\rho1}\rho), & \rho < r_1 \\ x_{2n}J_n(k_{\rho2}\rho) + x_{3n}Y_n(k_{\rho2}\rho), & r_1 < \rho < r_2 \\ x_{4n}J_n(k_{\rho3}\rho) + x_{5n}Y_n(k_{\rho3}\rho), & r_2 < \rho < r_3 \\ J_n(k_{\rho0}\rho) + c_n^{TM}H_n^{(2)}(k_{\rho0}\rho), & \rho > r_3 \end{cases} \quad (1)$$

$$H_\phi = E_{z0}e^{jk_{zi}z}\sum_n j^{-(n+1)}e^{jn\phi}\begin{cases} \left(\frac{k_1}{k_{\rho1}\eta_1}\right)x_{1n}J'_n(k_{\rho1}\rho), & \rho < r_1 \\ \left(\frac{k_2}{k_{\rho2}\eta_2}\right)[x_{2n}J'_n(k_{\rho2}\rho) + x_{3n}Y'_n(k_{\rho2}\rho)], & r_1 < \rho < r_2 \\ \left(\frac{k_3}{k_{\rho3}\eta_3}\right)[x_{4n}J'_n(k_{\rho3}\rho) + x_{5n}Y'_n(k_{\rho3}\rho)], & r_2 < \rho < r_3 \\ \left(\frac{k_0}{k_{\rho0}\eta_0}\right)[J'_n(k_{\rho0}\rho) + c_n^{TM}H'^{(2)}_n(k_{\rho0}\rho)], & \rho > r_3 \end{cases}$$
$$(2)$$

here, $E_{z0} = E_0 \sin\theta_i$ is the incident electric field strength, $k_{zi} = k_0 \cos\theta_i$, $k_{\rho l} = \sqrt{k_l^2 - k_{zi}^2}$, and $\eta_l$ are the wavenumber components and wave impedance in each region $l$, respectively, where $l = 0$ corresponds to free space. In (1)-(2), $J_n(\xi)$ and $Y_n(\xi)$ are Bessel and Neumann functions of order $n$, and the Hankel function is defined as $H_n^{(2)}(\xi) = J_n(\xi) - jY_n(\xi)$. In (2),

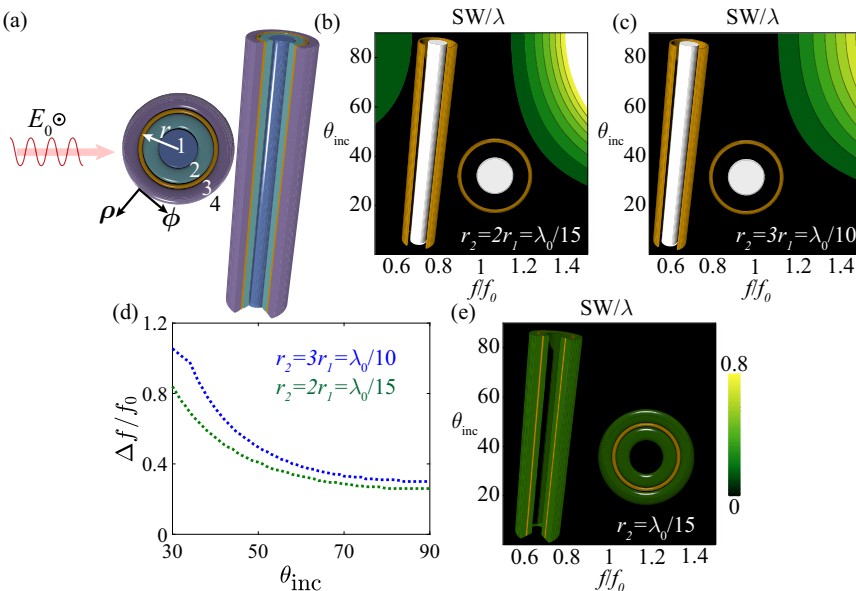

**Fig. 2 Scattering width (SW) of metasurface cloaks tailored to cloak conducting and dielectric cylinders. a** Schematic of a general multilayer infinitely long cylindrical structure. $E_0$ is the amplitude of the incident electric field. Panels **b** and **c** special cases of the design shown in **a**, where an infinitely long conducting cylinder is cloaked with an impedance metasurface. For **b**, the radius of the impedance sheet is twice the one of the metallic cylinder $r_2 = 2r_1 = \lambda_0/15$ with air as the spacer, and the optimal sheet impedance of the cloak is $Z_s = -j0.28\eta_0$. For the design in panel **c** the radius of the impedance sheet is three times the one of the metallic cylinder $r_2 = 3r_1 = \lambda_0/10$ with air as the spacer, and the sheet impedance is $Z_s = -j0.54\eta_0$. **d** Bandwidth ($\Delta f$) comparison between the cases shown in **b** and **c**. Here, the bandwidth is defined as the frequency band over which the normalized SW is <0.05. **e** Cloaking of a hollow dielectric cylinder. For this design, the radius of the impedance sheet is $r_2 = \lambda_0/15$, the thicknesses of super- and substrates are $\lambda_0/60$ and the sheet impedance is $Z_s = j1.39\eta_0$.

$\psi'_n(\xi) = (d/d\xi)\psi_n(\xi)$. At each interface, we apply the continuity of the tangential fields and the impedance boundary condition, yielding a rank-6 determinant. While not considered in this work, the surface impedance is in general dyadic $\overline{\overline{Z}}(\phi) = Z_{zz}\hat{z}\hat{z} + Z_{z\phi}\hat{z}\hat{\phi} + Z_{\phi\phi}\hat{\phi}\hat{\phi} + Z_{\phi z}\hat{\phi}\hat{z}$, where $Z_{z\phi} = Z_{\phi z} = 0$ and $Z_{zz} = Z_{\phi\phi} \neq 0$ reduces to the isotropic case considered here, such that $E_z(r_2^+) = E(r_2^-) = Z_{zz}[H_\phi(r_2^+) - H_\phi(r_2^-)]$[45]. Polarization coupling may be accounted for by considering the terms $\{Z_{z\phi}, Z_{\phi z}\}$, which may become detrimental or irrelevant depending on the cloak topology. By design, scalar surface impedance covers can enable minimal polarization coupling, leading to well-behaved and large scattering suppression bandwidths. Recently, single and bi-layer surfaces have been employed to reduce the antenna scattering to nearly zero, but at the cost of bandwidth reduction and potential polarization coupling between TM and TE modes[34,46]. Our goal here is to design the simplest cloaking structure that provides broadband and angle invariant scattering reduction, and yet preserves optimal radiation features.

As mentioned above, conventional cloaking approaches have been explored to suppress the scattering from low-frequency antennas over the entire high-frequency band. However, several issues limit the applicability of conventional cloaking technology to resonant conductive antennas. The most important challenge is the tradeoff between operational bandwidth and cloak thickness: because of the conductive nature of the antenna to be cloaked, a too narrow gap between metasurface cloak and antenna implies a large sensitivity to the operation frequency, making the cloak ineffective. At the same time, thicker cloaks are not only undesirable due to footprint considerations, but their response also becomes angular dependent, making the scattering reduction very sensitive to the location of the neighboring antennas. As an

illustrative example, Fig. 2b, c compare the normalized scattering width (SW) for conventional cloak designs with two different thicknesses, defined as $\sigma_{2D}/\lambda = \frac{2}{\pi \sin \theta_i}\sum_{n=0}^{N_{\max}}(2 - \delta_{0n})\left|c_n^{TM}\right|^2$ where $\delta_{0n}$ is the Kronecker discrete delta function, $\lambda$ is the free-space wavelength, and $N_{\max}$ is the maximum relevant order (determined by comparing the scattering coefficient of the $N_{\max}$ mode to that one of the dominant modes. Here, $N_{\max} = 9$), as a function of frequency and angle for a metallic cylinder cloaked with an optimal impedance sheet $Z_s$. With reference to Fig. 2a, for the examples shown in Fig. 2b, c, region 1 is a perfect electric conductor, and $\varepsilon_2 = \varepsilon_3 = \varepsilon_4 = 1$. An ideal impedance sheet is located at the boundary between regions 2 and 3 (i.e., $\rho = r_2$). The surface impedance value is numerically calculated using the procedure explained above with the goal of nullifying the scattering for the normally incident wave for two different radii of the impedance sheet. For the case shown in Fig. 2b, $r_2 = 2r_1 = \lambda_0/15$ and $Z_s = -j0.28\eta_0$, and for the design considered in Fig. 2c, $r_2 = 3r_1 = \lambda_0/10$ and $Z_s = -j0.54\eta_0$, where $\lambda_0$ is the wavelength of the excitation (the results in Fig. 2 are presented with dimensions normalized to the design wavelength $\lambda_0$ and all frequencies are normalized to $f_0 = c/\lambda_0$ where c is the speed of light). Since we aim at suppressing the influence of the cloaked antenna on high-frequency antennas placed in the near-field, the scattering suppression needs to be effective over a wide range of incidence angles. As expected, a wider frequency bandwidth is possible only at the cost of thicker designs, producing increased bandwidth sensitivity (defined as $\partial\left[\frac{\Delta f}{f_0}\right]/\partial\theta_{inc}$) to the illumination angle [see Fig. 2d]. The reason for this trade-off lies in the fact that the object we are trying to cloak is a strong scatterer to start with, typically a conductive dipole designed to efficiently radiate at a nearby frequency from the one in which we aim at suppressing the scattering.

To address this issue, we exploit a conceptually different approach to antenna cloaking, designing our metasurface for a subwavelength dielectric scatterer that on its own would be a very poor radiator. The inset in Fig. 2e shows the schematic of our cloaked object, consisting of a dielectric hollow core replacing the high-scattering metallic cylinder. Applying the boundary conditions explained above, a suitably tailored cloaking metasurface, inductive in nature, is inserted within the dielectric object to suppress its (capacitive) scattering. For this design, the radius of the impedance sheet is $r_2 = \lambda_0/15$, the thicknesses of super- and substrates are $\lambda_0/60$ and the sheet impedance is $Z_s = j1.39\eta_0$. As it is seen from the results shown in Fig. 2e, this design offers a dramatic improvement in terms of frequency and angular bandwidth. Particularly for larger angles, the dielectric object has very low scattering to start with, so the metasurface only needs to suppress the residual scattering for close-to-normal incidence, a much more relaxed goal than in the previous design. Here, the angular sensitivity is drastically reduced due to the metasurface being immersed inside the dielectric[47]. Since the uncoated dielectric cylinder has a relatively low scattering, the metsaurface cloak can be designed to efficiently suppress this scattering, and yet be highly insensitive to the angle of illumination and support broad bandwidths. At the same time, the implemented metasurface cloak, due to its conducting nature, can be optimized to be impedance matched to a regular feeding line and used itself as a radiating element. In this configuration, the dielectric core thus serves three purposes: it acts as a contrast element to eliminate the scattering of the metasurface, it offers mechanical support, and it capacitively enhances the coupling of the metasurface with the feeding source.

The radiating behavior of the immersed metasurface antenna is as equally important as the scattering cancellation behavior. Ideally, the radiating metasurface must support pure dipolar radiation without beam squinting. To realize the required impedance sheet, one way is to utilize helix designs; however, such designs usually suffer from narrow LB matching performance, beam squint, and cross-polarization issues[1]. In contrast, here we consider a simple inductive surface for dominant TM-polarization; i.e., a metasurface formed by metallic strips aligned with the incident wavefront, yielding an effective impedance[47]

$$Z_{strips}^{TM} = j\omega\frac{\mu_0 D}{2\pi}\ln\left[\csc\left(\frac{\pi w}{2D}\right)\right]\left(1 - \frac{\cos^2\theta_i}{2\varepsilon_{eff}}\right). \quad (3)$$

This simple but accurate formula calculates the effective shunt inductance for a given infinite planar surface, with strip width $w$, period $D$ and $\varepsilon_{eff} = (\varepsilon_l + \varepsilon_{l-1})/2\varepsilon_0$. It is well-known that thin, angularly stable surfaces for various applications are difficult to design while maintaining their functionality[47–49] and this intrinsic angular sensitivity is clearly seen in Eq. (3), as well. The design presented here provides much better polarization purity giving squint-less dipolar radiation with optimal beamwidth, and it is easier to match across a large bandwidth using a planar stub and a $\lambda/4$ transformer, implemented on a 3-layer board as typically done in modern antenna systems. The trace length and consequently the dielectric length are fine-tuned from the ordinary $\lambda/2$ conductive dipole length to optimize radiation in the LB [see Fig. 3]. Given the small radius of the structure and the required geometrical parameters to achieve the required inductance, the metasurface is actually realized by using two parallel strips embedded on opposite sides of the cylinders, with more details discussed in the Supplementary Materials. It is worth stressing that our results confirm that we can derive the optimal strip width and distance using Eq. (3) to synthesize the required inductive response, despite the fact that this formula refers to a planarized periodic surface, and not a curved one. Given that our

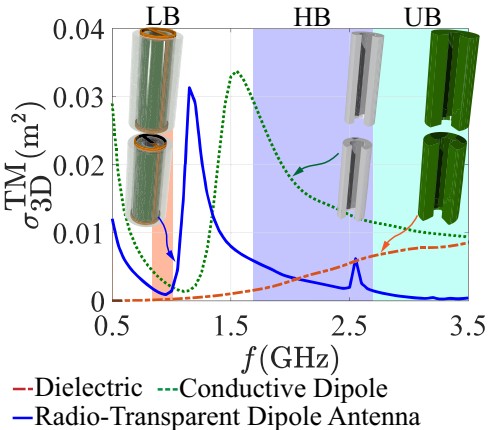

**Fig. 3 Scattering cross section of dipole antennas.** Total scattering cross section (SCS) for a normally incident plane wave across the lower band (LB), higher band (HB), and ultra-higher band (UB) for a conventional conductive dipole, a bare low-loss dielectric cylinder (without cloak) with $\varepsilon = \varepsilon_r(1 - j\tan\delta)$, where $\varepsilon_r = 4.4$ and $\tan\delta = 0.0005$, and for our radio-transparent antenna, all with same radii. The detailed dimensions and design details are given in Supplementary Fig. 3.

element is subwavelength, the quasi-static treatment used to derive Eq. (3) still works here, ensuring that the two strips provide the required overall inductive response necessary to optimally suppress the dielectric cylinder scattering.

In Fig. 3, we study the cloaking and scattering characteristics of our radio-transparent antenna based on this concept. Shown in the left inset of Fig. 3 is the design concept. The structure operates in the 698–968 MHz (LB) and suppresses the scattering in the 1.71–2.710 GHz (HB) range, since these bands are used for 3G and 4G LTE services. The frequency range around 3.5 GHz (UB) is also highlighted, as it is relevant in small cell or shared spectrum communications to alleviate the ever-growing wireless demands. The scattering features of our antenna are dominated by the electric dipole contribution, maximally excited by transverse magnetic (TM)-polarized waves at normal incidence, hence in this figure we analyze the scattering cross section for this excitation. The response for different incidence angles and different polarizations is shown in Supplementary Note 2, featuring negligible scattering similar to what was reported in Fig. 3. The figure compares the total scattering cross section (SCS) of a conventional conductive dipole, of a dielectric cylinder, and of our radio-transparent antenna. The conventional conductive dipole antenna, whose geometry is optimized to efficiently radiate in the LB (similar to a half-wavelength dipole), shows a significant SCS across the HB and UB, with a self-resonance around 1.5 GHz. The bare low-loss host dielectric (without embedded inductive cloak) with $\varepsilon = \varepsilon_r(1 - j\tan\delta)$, where $\varepsilon_r = 4.4$ and $\tan\delta = 0.0005$, has a much lower scattering, which grows in the upper frequency range as expected. The cloaked cylinder[30–33] is effective at suppressing the scattering in the HB and UB, and it also offers a pronounced scattering dip centered in the LB, of great interest to eliminate scattering interference with surrounding LB antennas in the array. The metasurface not only suppresses the scattering across the frequency range of interest compared to the dielectric rod (and much more if we compare it with the original conductive dipole), but it also offers a means to guide conduction currents that can radiate in the LB range. It is also important to stress that the very large bandwidth over which the scattering is suppressed in this design is enabled over a very compact footprint, with the cloak actually embedded within the dielectric cylinder, hence with zero contributions on the overall

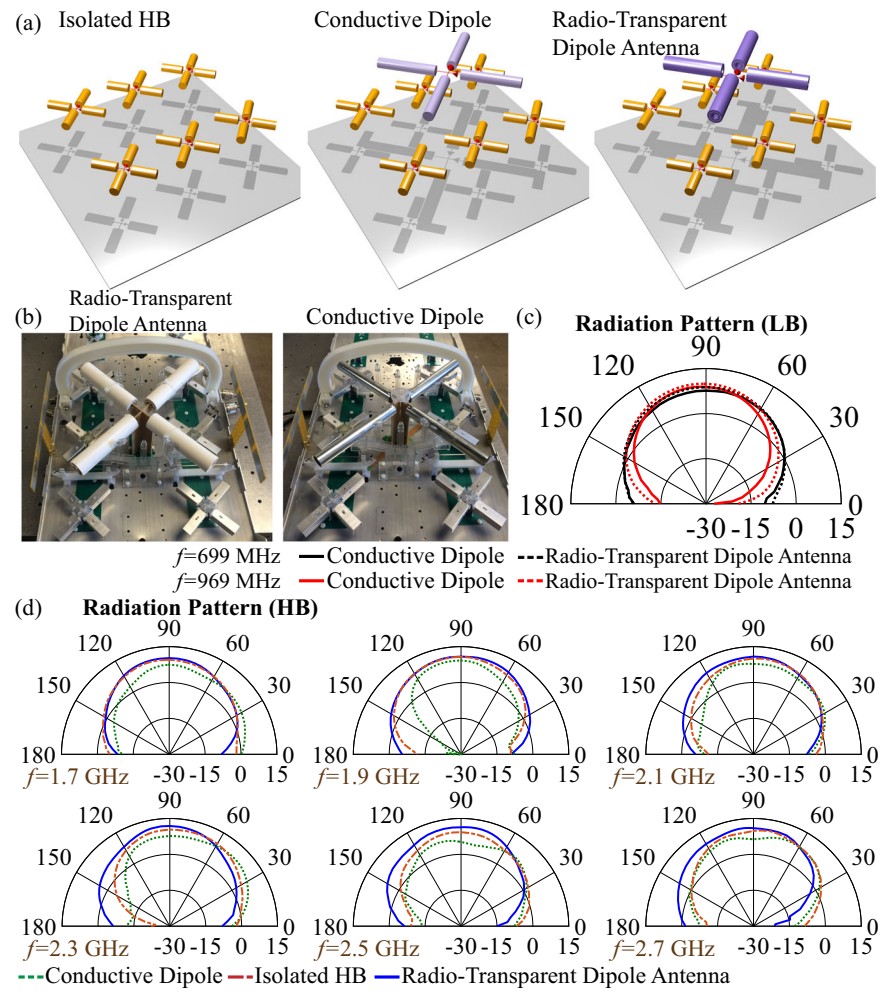

**Fig. 4 Experimentally measured radiation patterns. a** Schematic of the isolated higher band (HB) array, a typical communication system where a lower band (LB) conductive dipole is located over the HB array, and the proposed design where the radio-transparent antenna is located over the HB array. **b** Left panel: 2 × 3 panel unit cell of dual-polarized HB antenna elements with the proposed radio-transparent antenna above it. Note that, there is no phase shift between different polarizations and the antenna is dual polarized. Right panel: same unit cell with a conventional conductive dipole LB element. Each HB panel is designed to operate between 1.710 and 2.710 GHz (45% bandwidth), while the LB elements operate from 698 to 968 MHz (32% bandwidth). Far-field comparison across the **c** LB and **d** HB (experimental results). The detailed dimensions of the designs presented in this figure are given in the Supplementary Information. Note that reported radiation patterns already include the loss from different sources and all the mismatches, representing realized gain.

antenna footprint. This is different from any other approach for scattering suppression considered so far, which are typically unfeasible in terms of the resulting footprint, especially when bandwidth is a concern, and lead to increased scattering for TE-polarized wavefronts and other incidence angles due to an increased geometrical cross-section[27,35].

**Experimental demonstration**. We test the performance of our optimized cloak in terms of radiation features, exploring its LB performance in a conventional 3G and 4G LTE radio-base station, as shown in Fig. 4a. The figure compares three scenarios of interest: an isolated array of HB crossed dipoles, placed at a quarter wavelength from the ground plane; the dual-band array in which LB crossed dipoles are placed on top of the HB array, again at a quarter wavelength from the ground plane for their longer radiation wavelengths, and finally our radio-transparent antenna in a cross-dipole configuration replacing the LB conductive dipole. The LB elements are placed directly in front of the HB array because of the required distance from the ground, typically introducing a large disturbance in terms of beam squint, which

re-directs the HB radiation pattern away from its intended boresight direction ($\theta_i = 90°$). The beamwidth and gain of the HB elements can also be significantly altered by the presence of the LB elements. The total SCS of the LB element (conductive dipole antenna), as plotted in Fig. 3, is a good measure of its electromagnetic disturbance.

In our experiment, we considered a simplified 2 × 3 unit cell, which has non-ideal truncation effects including asymmetry and reduced ground plane effects [fabricated samples for the case of the conventional conductive dipole and the radio-transparent antenna are shown in Fig. 4b]. Additional details on each design and the testing methods are discussed in the Supplementary Note 3–7. Figure 4 showcases the improvement in terms of far-field scattering offered by our radio-transparent antenna across the entire operating bandwidth (LB and HB). In particular, Fig. 4c compares the LB performance between the conductive dipole and the radio-transparent antenna, while Fig. 4d compares the HB performance in the three scenarios of Fig. 4a, i.e., the conventional conductive dipole, the radio-transparent antenna, and the isolated scenario (removing the LB element). The LB radiation performance supports very good matching between the

conductive dipole and the radio-transparent antenna patterns. For both considered frequencies, the beamwidth becomes slightly narrower in the radio-transparent antenna, due to its slightly thicker overall radius when including the dielectric. Yet, the comparison clearly shows good dipolar radiation from the radio-transparent antenna, essentially mimicking the performance of the conventional conductive dipole.

The panels of Fig. 4d show the far-field performance at several frequencies across the HB. In almost all cases, the radiation from the radio-transparent antenna closely matches the one of the isolated panel, especially in terms of beam squint, gain and beamwidth. The LB conductive dipole, on the contrary, produces a strong squint the HB beam, essentially re-directing the beam away from boresight. In fact, at boresight we see strong blockage by the conductive dipole, reducing the HB gain by 4 dB across the band. The very low scattering of the radio-transparent antenna across the entire HB range is highly beneficial. We stress that there is no phase shift between different polarizations and the antenna system is dual-polarized. The far-field patterns in Fig. 4d demonstrate that the benefit of our radio-transparent antenna extends to both polarization planes. It should be noted that, here, we are considering a realistic practical design where there are several parasitic effects (e.g., coupling between the HB antennas, coupling between different HB antennas with the LB antenna, and the presence of a back reflector) that are not considered in the design process since it will introduce tremendous complications to the design process if not make it impossible. However, regardless of the complications mentioned, the proposed design performs incredibly well. By comparing the radiation patterns for the Conductive Dipole case and the proposed Radio-Transparent Dipole Antenna, it is clear that the proposed design performs extremely well at all HB frequencies by improving the radiation characteristics at these frequencies. However, ideally, we would want the proposed Radio-Transparent Dipole Antenna to match the radiation characteristics of the isolated HB antenna array. In fact, this is accomplished for lower frequencies as can be seen from the results. However, for higher frequencies $f \geq 2.3$ GHz, the radiation characteristics of the proposed Radio-Transparent Dipole Antenna are slightly deviated from the radiation characteristics of the Isolated HB antenna array which can be dedicated to the fact that, here, we are considering a system level proof of concept which has lots of extra parasitic effects that are extremely difficult if not impossible to take into account on the design level. It should also be noted that the proposed design improves the blockage issue almost in all the HB frequencies. From the results shown in Fig. 4d, it seems like the isolated antenna exhibits a lower gain on average than the cloaked one. This can be explained considering the fact that the antenna is not perfectly matched over the entire higher frequency band in the first place. The addition of the transparent antenna interacted with the HB array and improve its matching, hence resulting in an improved gain performance in comparison to the isolated case.

The far-field patterns in Fig. 4d already demonstrate improvements on the HB radiation patterns across a large bandwidth and angular spectra; however, in Fig. 5 we quantify the improvements offered by our approach using conventional performance metrics in commercial antenna systems. Beam squint and 3 dB beamwidth are the most challenging antenna metrics to restore when nearby obstacles introduce parasitic reflections. Up to the frequency of 1.91 GHz, both the conductive dipole and the radio-transparent antenna have little impact on the pattern in terms of beam squint. However, above this frequency, the conventional conductive dipole shows a strong deterioration on the HB radiation pattern. Yet, the radio-transparent antenna offers a remarkable field restoration across the whole band, with the squint becoming much lower and flatter. We emphasize that

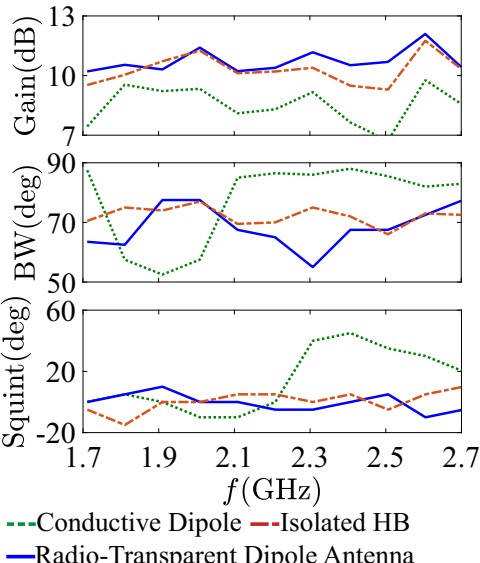

**Fig. 5 Radiation performance.** Measured metrics [gain, beamwidth (BW), and squint] for each testing case across the higher band (HB) frequency range.

the conductive dipole causes a re-direction of 20°–45° of the main beam between 2.310 and 2.710 GHz, while our cloaked antenna has a much lower range of re-directionality between 0°–5°, except for a narrowband 10° squint at 2.610 GHz. Across the entire HB, the average of the absolute values for beam squint caused by the conductive dipole element was measured to be 17.7° ± 16.8°, while for the cloaked antenna case this average was 4.2° ± 3.7°. For comparison, this measured average for the isolated panel was 5.2° ± 4.4°. It is also interesting to compare the measured beam squint to the total SCS calculated in Fig. 3, where we see the suppression bandwidth and the narrowband scattering peaks of the transparent dielectric core metasurface antenna. Below 2.0 GHz, the total SCS of the antenna increases, and a narrowband peak is noticed around 2.6 GHz. This peak is related to the angular stability of our design, which was minimized by using a simple inductive strip screen and is confirmed in our far-field measurements.

We have also studied the antenna gain and beamwidth, for which the conductive dipole introduces strong beamwidth instability across the band with a reduced gain. The average gain in the presence of the LB conductive dipole was measured to be 8.6 ± 1.9 dB. Meanwhile, the antenna cloak gain average was 11.9 ± 2.5 dB, and the average measured gain of the isolated case was 10.1 ± 2.9 dB. The average measured beamwidth for the conductive dipole was 77.4° ± 14.0°, radio-transparent dipole antenna was 68.5° ± 7.2°, and the isolated case was 72.2° ± 3.1°. When considering the gain and beamwidth, it is important to consider that they are measured at the maximum beam angle. Therefore, we must consider that the radio-transparent antenna and isolated scenarios are measured nearly at boresight, while the conductive dipole metrics are significantly skewed by the beam squint they introduce. By comparing these performance metrics holistically, the field restoration of the radio-transparent dipole antenna design is impressive.

In conclusion, we have introduced here broadband and efficient radio-transparent antennas of critical importance for modern communication systems (3G, 4G, and 5G), as more and more antennas need to be integrated on the same platform with minimal mutual interference. In recently explored approaches to enable low-scattering antennas, a conductive cylindrical antenna

forms the base of the radiating element, which is then covered with a metasurface to suppress its scattering over other frequency bands in which adjacent antennas operate. However, in these approaches the designed cloak typically works over a limited range of polarizations, incidence angles, and frequencies. Here, we explored a technique to enable broadband, dual-polarized radio-transparent antennas by exploiting the naturally low scattering of subwavelength dielectric objects. In this approach, the cloak not only suppresses undesired scattering from the dielectric core over a broad range of frequencies, but it also acts as an efficient radiator over the entire band of interest for transmit and receive operation. Such radio-transparent radiating elements can be of tremendous use in modern communication systems where more and more antennas need to be integrated into the same platforms. The proposed radio-transparent dielectric core metasurface antenna not only overcomes longstanding issues of conventional cloaking techniques in terms of narrow bandwidth, low angular stability, and sensitivity to the polarization of the incident wave, but it also shows that the cloak can act as an effective radiating element. Here, for proof of concept we applied this technique to antenna systems for 3G and 4G services; however, the same concept can be extended to 5G services that utilize several closely located frequency bands on the same platform, and to arbitrarily arranged cylindrical geometries (another example of applying this concept is provided in the Supplementary Note 8 demonstrating a transparent Yagi Uda antenna). However, it should be noted that the concept itself can be applied to any other geometries, including planar or spherical geometries by extending our theoretical formulation, as already explored for other mantle cloaking applications[30].

## Data availability
Data are available upon reasonable request from the authors.

## Code availability
Codes are available upon reasonable request from the authors.

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

## Acknowledgements
This work was partially supported by the Air Force Office of Scientific Research and the National Science Foundation.

## Author contributions
A.A. conceived the idea, J.S., Y.R., D.F., and A.A. developed the theoretical model. J.S., D.F., and Y.R. simulated and optimized the structure, built, and characterized the antenna. All authors interpreted the data and contributed to the manuscript preparation.

## Competing interests
The authors declare no competing interests.
