## [Peer Review File · Nature Communications]

REVIEWER COMMENTS

Reviewer #1 (Remarks to the Author):

The paper proposes a means for solving blockage and interference in dual-band closely-positioned antenna scenarios, e.g., as used in 3G and 4G cellular base stations. Typically, the low frequency (LB, large dimensions) antennas are positioned in front of the high frequency (HB, small dimensions) antennas in such a typical configuration, distorting the radiation patterns of the latter. A possible solution that has been attempted previously by numerous authors utilizes mantle cloaks (impedance surfaces) surrounding the LB antennas to reduce their scattering cross section at the high frequency band, allowing HB antennas to radiate properly. However, cloaking the LB antennas is typically narrow band, due to the resonant nature of the optimized conducting dipole radiators (alternatively, thicker cloaks can be used, which makes the overall design bulkier and less attractive). Instead, the authors propose to use the conducting impedance sheets comprising the cloak as the radiating element, and optimize them (together with a suitable dielectric core and cover) to exhibit low scattering cross section to begin with. From another point of view (as it is presented in the paper), instead of cloaking a resonant conductor, the idea is to cloak a dielectric cylinder (which is far less demanding since it has a significantly lower scattering cross section to begin with), and use the cloak surface as the means to conduct radiating currents required for an effective antenna. The concept is first principally discussed and illustrated using a simplified configuration, and then demonstrated experimentally using a commercially oriented base station antenna panel.

From a technical perspective, the work seems highly valuable. Indeed, it manages to solve the crowded-antennas problem in an elegant manner, maintaining the gain and beamwidth of the isolated HB antennas across the entire high frequency band even in the presence of the "blocking" (now cloaked) LB antenna, while avoiding undesired beam squint. At the same time, the proposed LB antenna, having a dielectric core and the cloak surface as the "active" element, performs as expected across the entire low frequency band. This may prove very useful for many modern (and future) communication systems. However, I have some concerns with regards to the paper as a scientific contribution intended for Nature Communications.

1. The immersed cloak design, which should have been the focus of this work, barely gets any attention at all. Besides being discussed mainly in the SM, the design formula is not even mentioned, there is no schematics of the surface itself, there is no schematics of the excitation circuit, no real discussion on how to "transform" the impedance surface into a commercial-grade antenna (besides one or two sentences on page 11 of the SM), no details on how it was fabricated. I think that the paper should be rewritten with these aspects receiving much more attention in the main text (including the discussion around Figs. S1 and S2). Even though the standard mantle cloak formulas may be well known, I think that the main design formulas, principal design considerations and guidelines, as well as the aforementioned schematics should appear in the main text.

2. Since the idea to use the cloak as the antenna has been proposed before (see [27] of the manuscript and [14] of the SM), it seems (the way the manuscript is written) that the main purpose of the work is to adapt this idea as to fit the commercial base station antenna panel scenario. While this makes the work very strong from a technical perspective, I think that adding another example (the authors mention "arbitrarily shaped antennas" on the last page) to demonstrate the universal nature of their concept (even in simulations) would significantly strengthen the scientific merit of the work.

3. Many details are missing or hard to locate, not all terms and acronyms are well defined, and terminology is not always consistent. All of these make the reading quite difficult.

3.1. Please introduce section numbers to the SM, and indicate these when making references in the main text.

3.2. Please identify references in the SM differently (maybe [S1], [S2], etc.) to avoid confusion.

3.3. In my mind, all the main definitions should appear in the main text (sending the reader to the SM to look for the definition of the scattering width does not seem reasonable, for instance).

3.4. Please define coordinate systems and polarizations (TE, TM) visually in the main text figure(s).

- 3.5. What is f_0 ? λ_0 ?
- 3.6. Main text refers to 3×3 unit cell for the antenna panel, while SM refers to 2×3 unit cell.
- 3.7. Page 4 – “a metallic cylinder cloaked with an optimal metasurface” – provide reference or a detailed design procedure (SM) to unambiguously identify the optimal metasurface.
- 3.8. Page 5 – “conducting dipole designed to efficiently radiate” – how does this design manifest itself in Fig. 2(a) and 2(b)?
- 3.9. Page 5 – [42] is referenced with respect to angular sensitivity, but we failed to locate where this issue is discussed therein.
- 3.10. Page 6 – “geometry is optimized to efficiently radiate in the LB” – how was the optimization performed? With respect to which figure of merit?
- 3.11. Page 15 – Please define the bandwidth mentioned in the caption. Is it the -3 dB bandwidth?
- 3.12. Page 17 – Please indicate what is plotted in the radiation patterns. Is it gain? Realized gain?
- 3.13. “SoM” and “SM” are inconsistently used to indicate Supplementary Material.
- 3.14. “Std. Dipole” and “CD” are inconsistently used to indicate conventional conducting dipole antennas.
- 3.15. Define TDCMA.

4. The authors should be clearer and more accurate regarding their description of performance improvement.

- 4.1. Page 6 – the authors mention increased geometrical cross-section of other solutions, referring to [33]. However, the cloak therein is only $0.063 \times \lambda_0$, substantially smaller than the one presented here.
- 4.2. Page 8 – the authors mention that in all cases in Fig. 4(d) the TDCMA closely matches the isolated panel. But there are significant differences in all the high-frequency ($f \geq 2.3$ GHz) patterns.

Other comments:

1. Page 4 – the authors mention increased sensitivity to illumination angle observed in Fig. 2(c). However, it seems that the band is wider for all angles of incidence for the thicker device...
2. Figure 2 – The perforated screen illustration is misleading, in my mind. If ideal impedance sheets were used, the illustration should include a smooth surface.
3. Page 6 – The authors mention that the (cloaked) LB antenna eliminates scattering interference also within the LB band. Could you please elaborate on how this is consistent with the bounds found by the authors in “Physical bounds on absorption and scattering for cloaked sensors” [Phys. Rev. B 89, 045122. 2014]? From reciprocity, if the antenna radiates well doesn't it mean it should also absorb well in the same frequency?
4. Page 7 – “an isolate array” (not “arrays”).
5. Page 8 – the authors claim that Fig. 4 shows the benefits of the proposed antenna for both polarization planes. How can this be seen? In which polarization were the measurements conducted? Which plane cut is shown?
6. Page 9 – Beam squint should be averaged in absolute value (cannot be negative). Otherwise large angular deviations with opposite signs would cancel each other.
7. Page 9 – How come the isolated antenna exhibits a lower gain on average than the cloaked one? Please comment on that in the manuscript.
8. SM, page 1 – don't x_1 , x_2 , etc. also depend on the Bessel function order “ n ”?
- 4.3. Please define coordinate systems and polarizations (TE, TM) visually in the SM figure(s).
9. SM, page 4 – how to determine the maximum relevant order N_{\max} ?
10. SM, page 4 – how Z_{zz} can be considered isotropic?
11. SM, Figure S6 – legend is inconsistent with the description in the text.

Reviewer #3 (Remarks to the Author):

The paper represents novel ideas that provide good information for current research but also ideas for advancing these ideas for future work.

It would be good to have more details on the actual experimental fabrication of the cloaking device and measurements. Confirmation of the dielectric constants of the materials used. Info on repeatability of the

experimental data etc.

Reviewer #4 (Remarks to the Author):

In the manuscript entitled "Radio-Transparent Dipole Antenna Based on a Metasurface Cloak", the authors proposed a radio-transparent antenna composed of a hollow dielectric cylinder and an inductive metasurface inserted within the dielectric cylinder. The inductive metasurface can suppress the total scattering of this antenna and meanwhile function as a radiating element, rendering it a radio-transparent antenna. This idea is a follow-up work of the previous concepts proposed by the authors as cloaked sensors, but the potential application in antennas seems quite interesting to me.

However, the manuscript seems not so satisfying in its current form and some important issues need to be addressed.

1. The idea of constructing a low-scattering antenna by a dielectric cylinder and an inductive metasurface inserted within the cylinder seems similar to the so-called "Mantle cloaking", which can cloak a dielectric cylinder by metasurface, except that the inductive metasurface is optimized to a feeding line. Considering this, the connections and differences between these two approaches should be discussed.
2. I wonder if the authors could directly compare with previous metal antennas with metasurface cloaks and scattering cancellation techniques, e.g. in Figs. 3,4 and 5.
3. What is the purpose of using the hollow cylinder instead of a solid one? How did the authors find this specific type of antenna? More elaboration on the reasoning behind the final result is preferred.
4. How important is the goal of cancelling the scattering of neighboring antennas in practice? I mean, if one looks at Fig. 1, the radiation pattern is only slightly affected by the low-frequency antenna. Thus this figure seems to indicate the scattering between neighboring antennas is just a minor issue, if one does not care too much about wavefront. Will this issue be significant for antenna array, or is there any circumstance in which this issue becomes critical?
5. The figures are not very attracting. For example, Figs. 2(a) and 2(b) seem to be the duplicate of each other, except for the change in the parameters. So why not plot only one figure and mark two sets of parameters to save the space? The font size in Fig. 3 are much larger than those in other figures.
6. As for the far-field performance at HB shown in Fig. 4(d), the improvement of the radiation from the radio-transparent antenna seems not so large. Especially, at 2.3 and 2.7GHz, the radiation of std. dipole seems more close to that of the isolated HB antenna. I suggest the authors add some analysis on the upper limit of the improvement and possibly negative factors that should be avoided.
7. The large and small antennas in Figure 4 are oriented to the same direction, what happens if they are not aligned to the same direction?
8. The curves in Figure 5 are not very smooth, probably due to insufficient frequency points.
9. Minor issues:
 - a. It seems that figures 4a and 4b show a 2x3 unit cell instead of a 3x3 one.
 - b. On page 8, in the second paragraph, "Figs. 4(d)" should be "Fig. 4(d)".

Overall, I find the concept and theory interesting, but the simulation and experimental results, as well as the presentation need to be substantially improved for publication on the high impact journal like nature communications.

**Response letter to the referees' comments on the manuscript NCOMMS-21-06952-T
entitled "Radio-Transparent Dipole Antenna Based on a Metasurface Cloak"**

We thank the Referees and the Editors for the time they spent evaluating our paper, for their constructive suggestions, and their overall positive feedback. We are very encouraged by the positive statements by all reviewers. In the following, we provide detailed response to all their comments, and outline the changes we implemented in the manuscript to address them. We hope that the revised manuscript may meet in the eyes of the Editors and the Referees the publication criteria of Nature Communications.

Referee A: The paper proposes a means for solving blockage and interference in dual-band closely-positioned antenna scenarios, e.g., as used in 3G and 4G cellular base stations. Typically, the low frequency (LB, large dimensions) antennas are positioned in front of the high frequency (HB, small dimensions) antennas in such a typical configuration, distorting the radiation patterns of the latter. A possible solution that has been attempted previously by numerous authors utilizes mantle cloaks (impedance surfaces) surrounding the LB antennas to reduce their scattering cross section at the high frequency band, allowing HB antennas to radiate properly. However, cloaking the LB antennas is typically narrow band, due to the resonant nature of the optimized conducting dipole radiators (alternatively, thicker cloaks can be used, which makes the overall design bulkier and less attractive). Instead, the authors propose to use the conducting impedance sheets comprising the cloak as the radiating element, and optimize them (together with a suitable dielectric core and cover) to exhibit low scattering cross section to begin with. From another point of view (as it is presented in the paper), instead of cloaking a resonant conductor, the idea is to cloak a dielectric cylinder (which is far less demanding since it has a significantly lower scattering cross section to begin with), and use the cloak surface as the means to conduct radiating currents required for an effective antenna. The concept is first principally discussed and illustrated using a simplified configuration, and then demonstrated experimentally using a commercially oriented base station antenna panel.

From a technical perspective, the work seems highly valuable. Indeed, it manages to solve the crowded-antennas problem in an elegant manner, maintaining the gain and beamwidth of the isolated HB antennas across the entire high frequency band even in the presence of the "blocking" (now cloaked) LB antenna, while avoiding undesired beam squint. At the same time, the proposed LB antenna, having a dielectric core and the cloak surface as the "active" element, performs as expected across the entire low frequency band. This may prove very useful for many modern (and future) communication systems. However, I have some concerns with regards to the paper as a scientific contribution intended for Nature Communications.

Authors: We thank the Referee for the solid summary of our paper, constructive comments, and positive feedback. We are happy that he/she has found the paper highly valuable. In the following, we address all his/her comments. We hope that the revised manuscript meets in the eyes of the Referee the publication criteria of Nature Communications.

***Referee A.1:** The immersed cloak design, which should have been the focus of this work, barely gets any attention at all. Besides being discussed mainly in the SM, the design formula is not even mentioned, there is no schematics of the surface itself, there is no schematics of the excitation circuit, no real discussion on how to "transform" the impedance surface into a commercial-grade antenna (besides one or two sentences on page 11 of the SM), no details on how it was fabricated. I think that the paper should be rewritten with these aspects receiving much more attention in the main text (including the discussion around Figs. S1 and S2). Even though the standard mantle cloak formulas may be well known, I think that the main design formulas, principal design considerations and guidelines, as well as the aforementioned schematics should appear in the main text.*

Authors: We agree with the referee that more details regarding the design and simulation procedure may make the paper more accessible for the reader. Following this comment, we have rewritten the paper and made sure that the revised version of the main paper includes the detailed design process of the cloak design. We also have revised some of the figures to help navigate through the design steps.

***Referee A.2.** Since the idea to use the cloak as the antenna has been proposed before (see [27] of the manuscript and [14] of the SM), it seems (the way the manuscript is written) that the main purpose of the work is to adapt this idea as to fit the commercial base station antenna panel scenario. While this makes the work very strong from a technical perspective, I think that adding another example (the authors mention "arbitrarily shaped antennas" on the last page) to demonstrate the universal nature of their concept (even in simulations) would significantly strengthen the scientific merit of the work.*

Authors: We thank the referee for this very useful suggestion. In the revised version of the supplementary materials, we have added another example of transparent antenna adapting this concept for a Yagi-Uda antenna, proving the general applicability of the proposed concept.

***Referee A.3.** Many details are missing or hard to locate, not all terms and acronyms are well defined, and terminology is not always consistent. All of these make the reading quite difficult.*

Authors: We agree with the referee that in the original submission these details were missing. In the revised version of the paper of the paper and SM, we have done our best to address these issues.

***Referee A.3.1.** Please introduce section numbers to the SM, and indicate these when making references in the main text.*

Authors: We thank the referee for pointing this out. To address this issue, we have introduced section numbers in the SM and use these numbers when citing specific parts of the SM.

***Referee A.3.2.** Please identify references in the SM differently (maybe [S1], [S2], etc.) to avoid confusion.*

Authors: We thank the referee for this comment. References now read as suggested by the referee.

Referee A.3.3. In my mind, all the main definitions should appear in the main text (sending the reader to the SM to look for the definition of the scattering width does not seem reasonable, for instance).

Authors: We thank the referee for this comment. In the revised version of the paper, the missing definitions have been added to the main text.

Referee A.3.4. Please define coordinate systems and polarizations (TE, TM) visually in the main text figure(s).

Authors: We thank the referee for this comment. In the revised version of the paper, the missing visual description has been added to Fig. 2. The rest of the figures in the main text and also SM already have the required visual description for the polarization of the incident wave.

Referee A.3.5. What is f_0 ? λ_0 ?

Authors: We thank the referee for pointing out the missing definitions. λ_0 is the wavelength of excitation. Note that, to keep our analysis general, in the results presented in Fig. 2 all the dimensions are normalized to the design wavelength λ_0 and all the frequencies are normalized to the design frequency $f_0 = c/\lambda_0$, where c is the speed of light. We have added an explanation in the revised version of the paper to clarify this.

Referee A.3.6. Main text refers to 3x3 unit cell for the antenna panel, while SM refers to 2x3 unit cell.

Authors: We thank the referee for noticing this misprint. The panel is a 2x3 unit cell. We have fixed the issue on the revised version of the paper.

Referee A.3.7. Page 4 - "a metallic cylinder cloaked with an optimal metasurface" - provide reference or a detailed design procedure (SM) to unambiguously identify the optimal metasurface.

Authors: We thank the referee for pointing out this issue and we agree that this could be a point of confusion for the readers. In the figures mentioned by the referee, the metallic cylinder is covered with an ideal impedance sheet. The required impedance can be found by applying the proper boundary conditions in Eqs. (1)-(2) of the main paper. We agree that in the original submission, the design procedure was not clear. To address this issue, in the revised paper we have added a detailed design procedure to prevent any confusion.

Referee A.3.8. Page 5 - "conducting dipole designed to efficiently radiate" - how does this design manifest itself in Fig. 2(a) and 2(b)?

Authors: We thank the referee for this comment. The design does not manifest itself in Fig. 2(a) or 2(b), since these figures assume an infinitely long cylinder and only consider their cloaking functionalities. Figure 4 is where we start discussing the radiative features of such designs.

Referee A.3.9. Page 5 - [42] is referenced with respect to angular sensitivity, but we failed to locate where this issue is discussed therein.

Authors: We thank the referee for this comment. The reference mentioned by the referee discusses the fact that the impedance of a metasurface when immersed in a dielectric has less angular sensitivity. However, to address this issue in the revised paper we added another reference where the same concept is discussed more clearly.

Referee A.3.10. Page 6 - "geometry is optimized to efficiently radiate in the LB" - how was the optimization performed? With respect to which figure of merit?

Authors: Here, the figure of merit is the reflection coefficient S_{11} and an optimization was performed such that a longitudinal resonance (close to half-wavelength dipole operation) was supported. To address this issue, in the revised paper we have reworded this portion of the text to make it clearer.

Referee A.3.11. Page 15 - Please define the bandwidth mentioned in the caption. Is it the -3 dB bandwidth?

Authors: Here, we define the bandwidth as the frequency range over which the normalized SW is less than 0.05. On the revised version of the paper, we have added this info to the caption of Fig. 2 to make it clearer.

Referee A.3.12. Page 17 - Please indicate what is plotted in the radiation patterns. Is it gain? Realized gain?

Authors: These results are all experimental results and we reported the radiation patterns already including the loss from different sources and all mismatches, representing the actual realized gain. We have added a note in the revised version of the paper to clarify this.

Referee A.3.13. "SoM" and "SM" are inconsistently used to indicate Supplementary Material.

Authors: We thank the referee for noting this inconsistency. In the revised version of the paper and SM, the text is consistently using SM.

Referee A.3.14. "Std. Dipole" and "CD" are inconsistently used to indicate conventional conducting dipole antennas.

Authors: We thank the referee for noting this inconsistency. In the revised version of the paper and SM, the text and figures are consistently using “conductive dipole”.

Referee A.3.15. Define TDCMA.

Reply: We thank the referee for noting the missing definition. In the revised version of the SM, we are now using a new abbreviation and have defined it there: Radio-Transparent Dipole Antenna Based on a Metasurface Cloak (RTDA).

Referee A.4. The authors should be clearer and more accurate regarding their description of performance improvement.

Referee A.4.1. Page 6 - the authors mention increased geometrical cross-section of other solutions, referring to [33]. However, the cloak therein is only $0.063 \cdot \lambda_0$, substantially smaller than the one presented here.

Authors: We thank the referee for this comment. The design presented here has a significantly reduced aspect ratio in comparison to the ones presented in the mentioned references by the referee. This is because of the added extra layers for cloaking in those works in contrast to an immersed cloaking layer that we have proposed in our work. For example: in [33] the total diameter of the monopole radiating at 2.4 GHz is 15.79 mm ($0.128 \lambda_0$ at 2.4GHz). The diameter of our proposed design is 22.88 mm which is $0.063 \lambda_0$ at a center BW of 830 MHz. Notice that this is the key advantage of our work: in conventional cloaked antennas, we cannot place the cloak too close to the metallic surface of the antenna without hindering the operational bandwidth. Here instead, we get rid of the metal, and use the cloak, immersed in a suitably designed dielectric, to radiate. We achieve therefore reduced cross-section and improved bandwidth at the same time.

Referee A.4.2. Page 8 - the authors mention that in all cases in Fig. 4(d) the TDCMA closely matches the isolated panel. But there are significant differences in all the high-frequency ($f \geq 2.3$ GHz) patterns.

Authors: We thank the referee for pointing out this important issue. To explain the performance of the proposed design, let us first start by considering a simple scenario where a LB antenna is illuminated at HB frequencies. Supplementary Fig. 8, shows that at all the tested HB frequencies, our proposed design performs extremely well in cloaking the LB antenna. However, in Fig. 4 of the paper we are considering a realistic and practical design where the LB antenna is positioned in front of an array of HB antennas and all are backed by a finite metallic back reflector rather than considering an abstract case where the LB antenna is located in front of one HB antenna. In such a practical scenario, there are several parasitic effects (e.g., coupling between the HB antennas, coupling between different HB antennas with the LB antenna, and the presence of a back reflector)

that are not considered in the design process since it will introduce tremendous complications to the design process, if not make it impossible. However, regardless of the complications mentioned, the proposed design performs quite well. There are a few notes that we need to consider when investigating the results in Fig. 4(d) of the paper. By comparing the radiation patterns for the Conductive Dipole case and the proposed Radio-Transparent Dipole Antenna, it is clear that the proposed design performs extremely well at all HB frequencies by improving the radiation characteristics at these frequencies. However, ideally, we would want the proposed Radio-Transparent Dipole Antenna to match the radiation characteristics of the isolated HB antenna array. In fact, this is accomplished for lower frequencies, as can be seen from the results.

As the referee correctly points out, for higher frequencies $f \geq 2.3$ GHz, the radiation characteristics of the proposed Radio-Transparent Dipole Antenna are slightly deviated from the radiation characteristics of the Isolated HB antenna array which can be explained by the fact that here we are considering a system level proof of concept which has lots of extra parasitic effects that are extremely difficult if not impossible to take into account carefully at the design level. It should also be noted that the proposed design improved the blockage issue almost in all the HB frequencies. To clarify the results more, we have added an explanation in the revised version of the paper.

Other comments:

Referee A.1. *Page 4 - the authors mention increased sensitivity to illumination angle observed in Fig. 2(c). However, it seems that the band is wider for all angles of incidence for the thicker device...*

Authors: We thank the referee for raising this issue. We apologize if our original statement was confusing. Basically, what we meant is that a thicker design will provide higher bandwidth but at the cost of *increased angular sensitivity* defined as $\partial \left[\frac{\Delta f}{f_0} \right] / \partial \theta_{inc}$. This can be seen from Fig. 2(d)

where the derivative of the blue line, corresponding to the thicker design, is sharper than the green one which corresponds to the thinner design. To clarify this, in the revised version of the paper we have added an explanation.

Referee A.2. *Figure 2 - The perforated screen illustration is misleading, in my mind. If ideal impedance sheets were used, the illustration should include a smooth surface.*

Authors: We thank the referee for the constructive comment. We agree with the referee that the original way of representing the impedance sheet could be misleading to the readers. In the revised version of the paper, we have addressed this issue by replacing the ideal impedance sheet with a smooth transparent surface and added a clarifying explanation to the text.

Referee A.3. *Page 6 - The authors mention that the (cloaked) LB antenna eliminates scattering interference also within the LB band. Could you please elaborate on how this is consistent with*

the bounds found by the authors in "Physical bounds on absorption and scattering for cloaked sensors" [Phys. Rev. B 89, 045122, 2014]? From reciprocity, if the antenna radiates well doesn't it mean it should also absorb well in the same frequency?

Authors: We thank the referee for bringing up this issue. Interestingly, and consistent with the results in the paper mentioned by this reviewer, a cloaked antenna can provide zero scattering and yet transmit / receive efficiently. The reason is that the absence of scattering from an antenna does not prevent the antenna from inducing currents on it, in reality the currents induced on our metal traces forming the antenna are responsible for cancelling the scattering from the dielectric host. Reciprocity requires that if we can induce currents on the metal traces, then by injecting those same currents we should be able to radiate out with the same efficiency, and indeed this is what we achieve. We have clarified this aspect in the main text.

Referee A.4. Page 7 - "an isolate array" (not "arrays").

Authors: We thank the referee for noticing this misprint. We have corrected the issue in the revised version of the paper.

Referee A.5. Page 8 - the authors claim that Fig. 4 shows the benefits of the proposed antenna for both polarization planes. How can this be seen? In which polarization were the measurements conducted? Which plane cut is shown?

Authors: The polarization of the received antenna is aligned with the length of the base station [vertical direction with respect to Fig. 4(b)]. With this in mind, the HB dipoles are diagonally oriented with respect to the receiving antenna, as commonly done in radio base station applications. Considering this and the fact that both arms of the HB antennas are excited simultaneously, the proof of concept considered here shows very good performance for both polarizations.

Referee A.6. Page 9 - Beam squint should be averaged in absolute value (cannot be negative). Otherwise large angular deviations with opposite signs would cancel each other.

Authors: We thank the referee for this comment. In this case, we did not use an averaged value but a single measurement at broadside when the HB antenna array is excited.

Referee A.7. Page 9 - How come the isolated antenna exhibits a lower gain on average than the cloaked one? Please comment on that in the manuscript.

Authors: We thank the referee for raising this issue. This could be explained considering the fact that the antenna cannot be perfectly matched over the entire higher frequency band (considering the simplicity of our design and matching network) in the first place. The addition of the transparent antenna interacted with the HB array and improved its matching, hence resulting in an improved gain performance in comparison to the isolated case. Furthermore, in such a practical

scenario, there are several parasitic effects (e.g., coupling between the HB antennas, coupling between different HB antennas with the LB antenna, and the presence of a back reflector) that are not considered in the design process since it will introduce tremendous complications.

Referee A.8. SM, page 1 - don't x_1 , x_2 , etc. also depend on the Bessel function order "n"?

Authors: We thank the referee for this comment. Yes, these numbers are dependent on the order n , the material parameters, geometry, excitation frequency, etc. In the design process, we aim to cancel the scattering only of the dominant mode (i.e. a single value of n). This procedure is valid because the scattering from a dipole antenna is largely dominated by the electric dipole contribution to the scattering cross section.

Referee A.4.3. Please define coordinate systems and polarizations (TE, TM) visually in the SM figure(s).

Authors: We thank the referee for this comment. Visual descriptions regarding the polarization of the incident waves have been added to the figures on supplementary material. We also have added direction of ρ and φ axes Figs. 1 and 2 and z and φ axes in Fig. 3 of the SM to represent the coordinate system.

Referee A.9. SM, page 4 - how to determine the maximum relevant order N_{max} ?

Authors: We thank the referee for this comment. The maximum order that is relevant is determined by comparing the scattering coefficient of the N_{max} mode to the one of the dominant mode. In this study, we included the first 10 modes. To address this, in the revised version of the paper, we have added a note to make it clearer.

Referee A.10. SM, page 4 - how Z_{zz} can be considered isotropic?

Authors: We thank the referee for pointing out this issue. Z_{zz} is just one tensor element of $\bar{\bar{Z}}$, so they are correct that the word 'isotropic' was not appropriate. In the revised SM, we have corrected the sentence.

Referee A.11. SM, Figure S6 - legend is inconsistent with the description in the text.

Reply: We thank the referee for noticing this misprint. In the revised version of the SM, the legend has been fixed.

Referee C. (Remarks to the Author):

The paper represents novel ideas that provide good information for current research but also ideas for advancing these ideas for future work.

Authors: We thank the referee for the constructive comments and positive feedback. We are happy that he/she has found the paper presenting novel ideas benefiting current and future work in this direction. In the following, we address all his/her comments. We hope that the revised manuscript may meet in the eyes of the Referee the publication criteria of Nature Communications.

Referee C.1. It would be good to have more details on the actual experimental fabrication of the cloaking device and measurements. Confirmation of the dielectric constants of the materials used. Info on repeatability of the experimental data etc.

Authors: We thank the reviewer for this comment. To address this issue, in the revised version of the supplementary materials (Supplementary Note 3), we have added the details on the fabrication of the antennas.

Referee D. (Remarks to the Author):

In the manuscript entitled "Radio-Transparent Dipole Antenna Based on a Metasurface Cloak", the authors proposed a radio-transparent antenna composed of a hollow dielectric cylinder and an inductive metasurface inserted within the dielectric cylinder. The inductive metasurface can suppress the total scattering of this antenna and meanwhile function as a radiating element, rendering it a radio-transparent antenna. This idea is a follow-up work of the previous concepts proposed by the authors as cloaked sensors, but the potential application in antennas seems quite interesting to me.

However, the manuscript seems not so satisfying in its current form and some important issues need to be addressed.

Authors: We thank the referee for the solid summary of our paper, constructive comments, and positive feedback. We are happy that he/she has found the paper quite interesting. In the following, we address all his/her comments. We hope that the revised manuscript may meet in the eyes of the Referee the publication criteria of Nature Communications.

Referee D.1. The idea of constructing a low-scattering antenna by a dielectric cylinder and an inductive metasurface inserted within the cylinder seems similar to the so-called "Mantle cloaking", which can cloak a dielectric cylinder by metasurface, except that the inductive metasurface is optimized to a feeding line. Considering this, the connections and differences between these two approaches should be discussed.

Authors: We thank the referee for raising this issue. The most common approach to implement a low scattering antenna is to start by designing an efficient LB antenna and then cloak it over the HB using different techniques, such as implementing an optimal mantle cloak. However, many of

the antennas designed based on this methodology not only suffer from narrow bandwidths and polarization sensitivity, but are also limited by the appendage of an extra cloaking increasing the aspect ratio. This is important in particular because the antenna to be cloaked is typically metallic, hence a cloaking metasurface needs to be placed at a distance to avoid shorting. Narrow gaps between the metasurface and the antenna lead to sacrificed bandwidth, whereas large gaps make the response angular dependent. Increased dimensions not only adversely affect the already closely packed system, but also result in undesired scattering through otherwise non-dominant scattering orders.

In this work, instead of cloaking an existing antenna, we start our design with a low scattering dielectric object and embed in it with an ultra-low profile metasurface to further reduce its scattering in the HB over a large bandwidth. This is easy to achieve, given the already low and non-resonant scattering of the dielectric host. We showed that this technique is capable of addressing the current challenges of cloaked antennas, realizing an efficient LB radiator and at the same time mitigating the total HB scattering of the element.

On the other hand, the approach presented here is similar to the conventional mantle cloak approach in the sense that they both are based on scattering cancellation concept. However, the impedance sheet used in our concept has a negligible cloaking responsibility since the dielectric has a very low scattering in the first place, making it less sensitive to the angle of illumination and provide higher bandwidth.

In the introduction of the paper, we already discuss the designs based on mantle cloaks and their challenges under the general concept of “scattering cancellation techniques” and mention their challenges. However, to make it clearer, we have added a discussion to the revised paper to reiterate the similarities and differences between the proposed design and with the ones based on mantle cloaks.

Referee D.2. I wonder if the authors could directly compare with previous metal antennas with metasurface cloaks and scattering cancellation techniques, e.g. in Figs. 3,4 and 5.

Authors: We thank the referee for this comment. We feel that this addition would be not particularly beneficial, as it is clear from the results shown in Fig. 2 that the scattering cross section of the immersed metasurface in a dielectric is notably reduced compared to the mantle cloak on a conductive dipole approach. In the interest of keeping our paper centered on our design, we prefer to avoid these additional plots. The challenge in drawing this comparison is that we improve both the cross section and the frequency / angular bandwidth of the response compared to conventional cloaked antennas. We have clarified these aspects further in the revised paper.

Referee D.3. What is the purpose of using the hollow cylinder instead of a solid one? How did the authors find this specific type of antenna? More elaboration on the reasoning behind the final result is preferred.

Authors: The center of the cylinder has been hollowed out for antenna mounting, weight reduction, as well as capacitive feeding for passive intermodulation reduction. Usually in real-life base stations the similar strategy is utilized for the same purposes. However, in principle, one can choose a solid cylinder if the above-mentioned factors are not important. In SM, we have a note explaining this.

Referee D.4. How important is the goal of cancelling the scattering of neighboring antennas in practice? I mean, if one looks at Fig. 1, the radiation pattern is only slightly affected by the low-frequency antenna. Thus this figure seems to indicate the scattering between neighboring antennas is just a minor issue, if one does not care too much about wavefront. Will this issue be significant for antenna array, or is there any circumstance in which this issue becomes critical?

Authors: We apologize if Fig. 1 of the paper in the original submission was misleading. In fact, Fig. 1 is a simplified schematic (and not a real simulated structure) for a real-life base station scenario. In the revised version of the paper, we have modified this figure to prevent confusion.

Referee D.5. The figures are not very attracting. For example, Figs. 2(a) and 2(b) seem to be the duplicate of each other, except for the change in the parameters. So why not plot only one figure and mark two sets of parameters to save the space? The font size in Fig. 3 are much larger than those in other figures.

Authors: We thank the referee for the comment. Figures 2(a) and 2(b) are two-dimensional intensity plots and, although we understand and appreciate the referee's point in making the figures more compact and appealing, unfortunately it is not possible to merge them into one plot. However, in the revised version of the paper, we have modified these figures to make them more attractive inspired by these suggestions.

Referee D.6. As for the far-field performance at HB shown in Fig. 4(d), the improvement of the radiation from the radio-transparent antenna seems not so large. Especially, at 2.3 and 2.7GHz, the radiation of std. dipole seems more close to that of the isolated HB antenna. I suggest the authors add some analysis on the upper limit of the improvement and possibly negative factors that should be avoided.

Authors: We thank the referee for this comment. To explain the performance of the proposed design, let us first start by considering a simple scenario where a LB antenna is illuminated at HB frequencies. Supplementary Fig. 8 shows that at all the tested HB frequencies the proposed design performs extremely well in cloaking the LB antenna. However, in Fig. 4 of the paper, we are considering a realistic practical design where the LB antenna is positioned in front of an array of HB antennas and all are backed by a metallic back reflector rather than considering an abstract case where the LB antenna is located in front of one HB antenna. In such a practical scenario, there are several parasitic effects (e.g., coupling between the HB antennas, coupling between different HB antennas with the LB antenna, and the presence of a back reflector) that are not considered in the design process since it will introduce tremendous complications to the design process if not

make it impossible. However, regardless of the complications mentioned, the proposed design performs quite well. There are a few notes that we need to consider when investigating the results in Fig. 4(d) of the paper. By comparing the radiation patterns for the Conductive Dipole case and the proposed Radio-Transparent Dipole Antenna, it is clear that the proposed design performs extremely well at all HB frequencies by improving the radiation characteristics at these frequencies. However, ideally, we would want the proposed Radio-Transparent Dipole Antenna to match the radiation characteristics of the isolated HB antenna array. In fact, this is accomplished for lower frequencies as can be seen from the results. However, as the referee correctly points out, for higher frequencies $f \geq 2.3$ GHz, the radiation characteristics of the proposed Radio-Transparent Dipole Antenna are slightly deviated from the radiation characteristics of the Isolated HB antenna array which can be dedicated to the fact that here we are considering a system level proof of concept which has lots of extra parasitic effects that are extremely difficult if not impossible to take into account on the design level. It should be noted that the proposed design improved the blockage issue almost in all the HB frequencies.

The significant improvement induced by the proposed design can also be seen from Fig. 5 of the main paper. For most of the band, the gain is fully restored, which goes directly into the power transmission of the base-station, especially since the antenna is directly in front of the transmitter, where every fraction of a dB is precious. Considering even a modest 1dB band averaged gain improvement, the base station would be 20% more efficient with the proposed approach. Most of the in-band gain improvement is ~ 3 dB, which is impressive. Next consider the additional degradation due to beam squint. Even if the base station was 100% efficient, for half of the band, the power would be directed 45deg. from its intended direction. The bandwidth is dramatically larger than any similar approach. To clarify the results more, we have added an explanation in the revised version of the paper.

Referee D.7. The large and small antennas in Figure 4 are oriented to the same direction, what happens if they are not aligned to the same direction?

Authors: We thank the referee for this comment. Although not considered in this paper, we believe that there should not be a significant change in the response. As it can be seen from Fig. 4, here half of the HB dipoles are aligned with the LB dipoles and the other half are orthogonal to them, showing the reliability of the design under the two extremes. Furthermore, it should be noted that here we are exciting all the dipoles simultaneously. A rotation of one dipole with respect to the other would not change significantly the results because of the small size of the antennas and the fact that we are exciting already both polarizations.

Referee D.8. The curves in Figure 5 are not very smooth, probably due to insufficient frequency points.

Authors: We thank the referee for this comment. We agree with the referee that the number of frequencies were not enough to create a smooth graph however we believe we have enough

frequency points in our measurements to properly capture the performance of the structure within the frequency range of interest.

Referee D.9. Minor issues:

Referee D.a. It seems that figures 4a and 4b show a 2x3 unit cell instead of a 3x3 one.

Authors: We thank the referee for noticing this misprint. Indeed, the unit cell is a 2x3 unit cell. On the revised version of the paper we have corrected this misprint.

Referee D.b. On page 8, in the second paragraph, "Figs. 4(d)" should be "Fig. 4(d)".

Authors: We thank the referee for noticing this misprint. In the revised version of the paper, we have corrected this misprint.

Overall, I find the concept and theory interesting, but the simulation and experimental results, as well as the presentation need to be substantially improved for publication on the high impact journal like nature communications.

Authors: We thank the referee for helping us improve drastically the quality of our presentation. We hope the revised paper may meet to their eyes the high standards of Nature Communications.

REVIEWER COMMENTS

Reviewer #1 (Remarks to the Author):

I thank the authors for the thorough consideration of my comments; most of them were addressed satisfactorily. However, further to additional clarifications required for some of the responses, one major issue has emerged.

Particularly, the additional details and improved schematics of the "metasurface cloak" design provided in the revised manuscript shed new light on the presented results and brought up new issues. In fact, as it became clear now from the improved drawings in Figure 3 and Supplementary Figures 3 and 4, the "metasurface" contains only two thin strips along the cylinder circumference. This is highly unconventional as far as metasurfaces are concerned (e.g., see the metasurface cloaks in [35], [37]). This aspect has to be clarified by the authors before I can make my recommendation, since it is central to the claims made in the manuscript.

1. Is such a sparse configuration sufficient for homogenization? Could this two-strip structure be considered an impedance surface? If the authors claim this is the case, they should provide direct evidence to that, by extracting in simulations the surface impedance of such a structure (regardless of the cloaking functionality), and comparing it to analytical models based on homogenization (e.g., [37]).
2. Looking again in Fig. 3, it is not obvious to me that the thin strips play an important role in the cloaking effect as described by the authors. To establish this, the authors should indicate clearly how this final configuration is obtained based on Eq. (3). Which frequency is used in this equation as the cloaking frequency that yields the implemented design parameters? Is the cloaking quality reduced when the thin strip width deviates from the derived value?
3. Could it be that the main scattering reduction stems from the use of the dielectrics, and the strips merely serve as means for establishing radiation (i.e., exciting the dielectric body)? In that sense, other conductor geometries (regardless of the metasurface cloak design rules) might be as good as these thin strips in providing similar performance.
4. Either way, the fact that two thin strips are considered a "metasurface" should be clearly highlighted and commented about in the main text (considering also the points mentioned above). A clear figure of the cloak itself (e.g., as in [35], [37]) should be presented, with all the relevant dimensions denoted on it. In the current manuscript form, this fact can be easily missed, and is mostly documented in the Supplementary Material (also not emphasized sufficiently therein).

Below I refer to additional revisions or clarifications required with regards to specific comments made in my previous report.

A.1: I advise the authors to provide in the future specific and detailed references to where revisions have been made in the text, which would surely expedite the review process. I may have missed it, but I could not find the schematics of the excitation circuit that was requested in this comment, as well as clear schematics of the metasurface itself. Please add these to the paper.

A.2: As far as I understand from the authors' response and provided example, "arbitrarily shaped antennas" in page 15 actually means "arbitrary arrangements of dipole antennas"? Please clarify or rephrase the statement.

A.3.2: I could not find the modified reference notations in the SM. Please verify.

A.3.12: Please specify also the polarization details of the measurements in the caption (see also Other comments/A.5).

A.4.1: Ok, thank you for the clarification. A minor issue - in Supp. Fig. 3 the value of r_3 is different than in Supp. Fig. 4. Please check.

A.4.2: Ok, thank you for clarifying. I think, however, that there was a confusion in the revised text, since the section "In such a practical scenario, there are several parasitic effects (e.g., coupling between the HB antennas, coupling between different HB antennas with the LB antenna, and the presence of a back reflector) that are not considered in the design process since it will introduce tremendous complications to the design process if not make it impossible." appears with an almost identical wording at least twice therein. Please check.

Other comments/A.5: These polarization aspects should be clarified in the manuscript, including in the caption of Fig. 4. Is there any phase shift between the crossed-dipole arms? Do the cross dipoles emit circularly polarized radiation?

Other comments/A.6: I'm not sure what the authors mean in their response. The text specifically reads "Across the entire HB, the average beam squint caused by the conductive dipole element was measured to be ..." (page 13), namely, there is a specific reference to averaging. Please clarify this in view of our comment.

Other comments/A.8: I may have not phrased my comment properly. Since these coefficients appear in the summation, it may be appropriate to indicate explicitly (using a subscript or a superscript) the fact that they are dependent on n .

Reviewer #4 (Remarks to the Author):

I read the revised manuscript and the response letter carefully. The authors have made a substantial revision to their manuscript. The current version is much better than the previous one. They have also nicely addressed all my comments. I still think this original work would have an important impact to the field of antennas and metamaterials. Thus I am happy to recommend for publication. I only have some minor issues as listed below:

a) In the revised manuscript, the authors mentioned that some metasurface cloak schemes have been applied to reduce antenna scattering. I'd like to suggest the authors add some brief review on metasurface cloak and transparent structures as well as some closely related references, so as to help the readers establish a more comprehensive understanding of the background of cloaking or transparent structures.

For example, some important works may include:

1. An ultrathin directional carpet cloak based on generalized Snell's law. *Appl. Phys. Lett.* 103, 151115 (2013).
2. An ultrathin invisibility skin cloak for visible light, *Science* 349, 1310 (2015).
3. Terahertz carpet cloak based on a ring resonator metasurface, *Phys. Rev. B* 91, 195444 (2015).
4. Ultratransparent Media and Transformation Optics with Shifted Spatial Dispersions, *Phys. Rev. Lett.* 117, 223901 (2016).
5. A hybrid invisibility cloak based on integration of transparent metasurfaces and zero-index materials, *Light Sci. Appl.* 7, 50 (2018).
6. Invisible surfaces enabled by the coalescence of anti-reflection and wavefront controllability in ultrathin metasurfaces, *Nat. Commun.* 12, 4523 (2021).

b) There are still some typos in the revised manuscript and supplementary materials. For example, in the figure caption of Supplementary Figure 11, the description on the (c) part is missing.

Response letter to the referees' comments on the manuscript NCOMMS-21-06952-T entitled "Radio-Transparent Dipole Antenna Based on a Metasurface Cloak"

We thank the Referees for the time they spent evaluating our paper, for their constructive suggestions, and their overall positive feedback. We are very encouraged by the positive statements by all reviewers. In the following, we provide detailed response to all their comments, and outline the changes we implemented in the manuscript to address them. We hope that the revised manuscript may meet in the eyes of the Editors and the Referees the publication criteria of Nature Communications.

Reviewer #1 (Remarks to the Author):

I thank the authors for the thorough consideration of my comments; most of them were addressed satisfactorily. However, further to additional clarifications required for some of the responses, one major issue has emerged.

Particularly, the additional details and improved schematics of the "metasurface cloak" design provided in the revised manuscript shed new light on the presented results and brought up new issues. In fact, as it became clear now from the improved drawings in Figure 3 and Supplementary Figures 3 and 4, the "metasurface" contains only two thin strips along the cylinder circumference. This is highly unconventional as far as metasurfaces are concerned (e.g., see the metasurface cloaks in [35], [37]). This aspect has to be clarified by the authors before I can make my recommendation, since it is central to the claims made in the manuscript.

1. Is such a sparse configuration sufficient for homogenization? Could this two-strip structure be considered an impedance surface? If the authors claim this is the case, they should provide direct evidence to that, by extracting in simulations the surface impedance of such a structure (regardless of the cloaking functionality) and comparing it to analytical models based on homogenization (e.g., [37]).

Authors: We thank the referee for raising this issue. The reviewer certainly raises a very important point. In agreement with this comment, it is not trivial to homogenize a curved surface, and it is not trivial to treat a pair of strips as a homogenized surface impedance. Interestingly, in our scenario the homogenization actually works very effectively, as we discuss in the following. Given that the strips are wrapped around a circular pattern, effectively our structure is periodic with a period of $0.28\lambda_0$. Based on our theoretical analysis based on satisfying boundary conditions, the required homogenized impedance to suppress the scattering at 3.5 GHz is $j243.6$ Ohms. Based on the analytical formula given in Eq. 3, the vertical strips with the selected width and periodicity used in Fig. 3 would provide an impedance $j249.8$ Ohms at the design frequency, very close to the required value. We calculated with full-wave simulations the effective surface impedance of a planarized metasurface with same strip geometry, i.e., a periodic array of infinitely long strips with same width and periodicity that we have used in our design. The following figure shows the real and imaginary parts of this surface impedance from full-wave simulations. It can be seen that at

the design frequency 3.5 GHz, the surface indeed provides a homogenized impedance $0.27 + j240.52 \text{ Ohm}$, which is very close to our requirement.

It is rather remarkable that the two strips around the antenna do provide the required impedance, but indeed this functionality can indeed be checked by looking at the SCS graph in Fig 3 and Supplementary Fig. 2 for the cloaked antenna. As expected from the design, the maximum scattering cancellation happens at the design frequency (3.5GHz). This is consistent with our past experience on metasurface cloaks, and it can be explained by the fact that the impedance surface realizing the mantle cloak is subwavelength, so the two strips provide the required effective inductance in a quasi-static scenario that is consistent with the homogenization procedure used to define an effective surface impedance for a planar metasurface with subwavelength periodicity, as in Eq. (3) of the main text. Our results indeed confirm this intuition quantitatively. We have clarified these important points in the revised paper.

2. Looking again in Fig. 3, it is not obvious to me that the thin strips play an important role in the cloaking effect as described by the authors. To establish this, the authors should indicate clearly how this final configuration is obtained based on Eq. (3). Which frequency is used in this equation as the cloaking frequency that yields the implemented design parameters? Is the cloaking quality reduced when the thin strip width deviates from the derived value?

Authors: We thank the referee for this comment. As mentioned in the response to the previous comment, we do obtain the design parameters for strips starting with the required surface impedance derived by applying the appropriate boundary conditions in Eqs. 1 and 2 (in our design

it is $j243.6$ Ohms). Then, we plug this required impedance value into Eq. 3 and find the periodicity and width of the strips. The design frequency was 3.5GHz in this case. And the answer to the referee's last question is yes. The cloaking effect deteriorates if we change the width of the strips, as expected.

3. Could it be that the main scattering reduction stems from the use of the dielectrics, and the strips merely serve as means for establishing radiation (i.e., exciting the dielectric body)? In that sense, other conductor geometries (regardless of the metasurface cloak design rules) might be as good as these thin strips in providing similar performance.

Authors: We thank the referee for raising this doubt. This is not the case. Here, the designed metasurface directly contributes to the reduction of the SCS of the host dielectric cylinder. This can be seen in Fig. 3, where maximum SCS reduction happens at the design frequency (i.e., 3.5GHz). At this design frequency, the SCS of the cloaked dielectric (blue curve in Fig. 3), i.e., metallic strips embedded inside the dielectric cylinder, is much less than that of the bare dielectric (red curve in Fig. 3).

To clarify this issue further, here, using the theoretical analysis in the main paper, we compare the coefficients of the dominant scattering mode as well as the scattering cross section for the infinitely long bare and cloaked dielectric cylinders. For this comparison, the metasurface was modeled using Eq. 3, since, as it was shown in response to the previous comments, it is safe and valid to use the effective impedance model in Eq. 3 to model the two-strip configurations used in our design. The following figures show that, at the design frequency (3.5 GHz), the coefficient for the dominant scattering mode becomes zero while for the bare dielectric this coefficient is not at all close to zero. Consequently, the scattering width of the cloaked dielectric cylinder is much lower, showing that indeed the cloak considerably reduces the scattering from the dielectric cylinder, in line with the discussion in the paper.

4. Either way, the fact that two thin strips are considered a “metasurface” should be clearly highlighted and commented about in the main text (considering also the points mentioned above). A clear figure of the cloak itself (e.g., as in [35], [37]) should be presented, with all the relevant dimensions denoted on it. In the current manuscript form, this fact can be easily missed, and is mostly documented in the Supplementary Material (also not emphasized sufficiently therein).

Authors: We thank the referee for the suggestion. In the revised paper, we have added an explanation in the main text (end of page 8) and in Supplementary Note 2 addressing the issue raised by the referee, specifically highlighting the fact that we can model the two strips as an inductive metasurface. Furthermore, we have added an inset in Supplementary Fig. 3 of the SM where we show the unwrapped metasurface with all the dimensions in it. We believe all dimensions are now given in this figure.

Below I refer to additional revisions or clarifications required with regards to specific comments made in my previous report.

*A.1: I advise the authors to provide in the future specific and detailed references to where revisions have been made in the text, which would surely expedite the review process. I may have missed it, but I could not find the schematics of the excitation circuit that was requested **in this** comment, as well as clear schematics of the metasurface itself. Please add these to the paper.*

Authors: We thank the referee for this suggestion. In this round of revision, we have added notes to each response on where the changes have happened to make it easier for the referee to track them. We also thank the referee for the note on the excitation circuit. In the revised SM, we have added the schematic of the excitation matching network for the cloaked dipole in Fig. 4 of the SM together with all the required dimensions in the caption of the figure. We also have added an explanation regarding the excitation matching network in the last paragraph of the page 7 in the revised SM.

A.2: As far as I understand from the authors’ response and provided example, “arbitrarily shaped antennas” in page 15 actually means “arbitrary arrangements of dipole antennas”? Please clarify or rephrase the statement.

Authors: We thank the referee for raising this point. In this paper we have developed the theory for cylindrical geometries. Therefore, as the referee correctly points out, our theory can be applied to any cylindrical shaped antenna, dipoles, monopoles, periodic log, and so on. An example of this is shown in the SM following one of the comments from the referee during the first round of the review, where the concept was applied to a Yagi Uda antenna. However, it should be stressed that the introduced concept itself can be applied also to totally different geometries. To address this issue, in the revised paper we have reworded the specified sentence by the referee in the conclusion at page 15 of the revised paper.

A.3.2: I could not find the modified reference notations in the SM. Please verify.

Authors: We thank the referee for noticing this misprint. In the revised version of the SM, we have corrected this issue.

A.3.12: Please specify also the polarization details of the measurements in the caption (see also Other comments/A.5).

Authors: We thank the referee for this suggestion. In the revised paper, we have addressed this comment (Also the comments/A.5).

A.4.1: Ok, thank you for the clarification. A minor issue - in Supp. Fig. 3 the value of r_3 is different than in Supp. Fig. 4. Please check.

Authors: We thank the referee for bringing this up for clarification. The fabricated sample came with a 5% error in r_3 value from the designed value. For this reason, we have a different value for r_3 in the fabricated sample. To clarify this, we have added a note in the caption of the Fig. 4 of the SM.

A.4.2: Ok, thank you for clarifying. I think, however, that there was a confusion in the revised text, since the section "In such a practical scenario, there are several parasitic effects (e.g., coupling between the HB antennas, coupling between different HB antennas with the LB antenna, and the presence of a back reflector) that are not considered in the design process since it will introduce tremendous complications to the design process if not make it impossible." appears with an almost identical wording at least twice therein. Please check.

Authors: We thank the referee for noticing this misprint. We have corrected the issue in the revised version of the paper.

Other comments/A.5: These polarization aspects should be clarified in the manuscript, including in the caption of Fig. 4. Is there any phase shift between the crossed-dipole arms? Do the cross dipoles emit circularly polarized radiation?

Authors: We thank the referee for bringing this up. There is no phase shift between different polarizations. The radiation is dual polarized and NOT circularly polarized. To clarify this issue, we have added an explanation in the revised paper in the caption of Fig. 4 and also in the last paragraph of page 11 in the revised paper.

Other comments/A.6: I'm not sure what the authors mean in their response. The text specifically reads "Across the entire HB, the average beam squint caused by the conductive dipole element was measured to be ..." (page 13), namely, there is a specific reference to averaging. Please clarify this in view of our comment.

Authors: We thank the referee for noticing this issue. We misunderstood the referee's earlier comment in the previous round of reviews. We completely agree with the referee that the average should be of the absolute value. We have redone the calculations and corrected it, as suggested by the referee, and rephrased the text in page 13 of the revised paper.

Other comments/A.8: I may have not phrased my comment properly. Since these coefficients appear in the summation, it may be appropriate to indicate explicitly (using a subscript or a superscript) the fact that they are dependent on n .

Authors: We thank the referee for the suggestion. Following this recommendation, we have modified the suggested scripts in Eqs. 1 and 2 in the revised paper.

Reviewer #4 (Remarks to the Author):

I read the revised manuscript and the response letter carefully. The authors have made a substantial revision to their manuscript. The current version is much better than the previous one. They have also nicely addressed all my comments. I still think this original work would have an important impact to the field of antennas and metamaterials. Thus I am happy to recommend for publication. I only have some minor issues as listed below:

a) In the revised manuscript, the authors mentioned that some metasurface cloak schemes have been applied to reduce antenna scattering. I'd like to suggest the authors add some brief review on metasurface cloak and transparent structures as well as some closely related references, so as to help the readers establish a more comprehensive understanding of the background of cloaking or transparent structures.

For example, some important works may include:

- 1. An ultrathin directional carpet cloak based on generalized Snell's law. Appl. Phys. Lett. 103, 151115 (2013).*
- 2. An ultrathin invisibility skin cloak for visible light, Science 349, 1310 (2015).*
- 3. Terahertz carpet cloak based on a ring resonator metasurface, Phys. Rev. B 91, 195444 (2015).*

4. *Ultratransparent Media and Transformation Optics with Shifted Spatial Dispersions*, *Phys. Rev. Lett.* 117, 223901 (2016).

5. *A hybrid invisibility cloak based on integration of transparent metasurfaces and zero-index materials*, *Light Sci. Appl.* 7, 50 (2018).

6. *Invisible surfaces enabled by the coalescence of anti-reflection and wavefront controllability in ultrathin metasurfaces*, *Nat. Commun.* 12, 4523 (2021).

Authors: We thank the referee for this suggestion. Following this recommendation, we have added some of the proposed references to the revised paper.

b) There are still some typos in the revised manuscript and supplementary materials. For example, in the figure caption of Supplementary Figure 11, the description on the (c) part is missing.

Authors: We thank the referee for noticing these misprints. We have done our best to address these misprints in the revised paper and SM.

REVIEWERS' COMMENTS

Reviewer #1 (Remarks to the Author):

I thank the authors once again for the thorough consideration of my comments. They have properly addressed all my concerns and I gladly recommend publication.